# The Selective α1 Antagonist Tamsulosin Alters ECM Distributions and Cellular Metabolic Functions of ARPE 19 Cells in a Concentration-Dependent Manner

**DOI:** 10.3390/bioengineering9100556

**Published:** 2022-10-14

**Authors:** Yosuke Ida, Tatsuya Sato, Megumi Watanabe, Araya Umetsu, Yuri Tsugeno, Masato Furuhashi, Fumihito Hikage, Hiroshi Ohguro

**Affiliations:** 1Department of Ophthalmology, Sapporo Medical University School of Medicine, Sapporo 060-8556, Japan; 2Department of Cardiovascular, Renal and Metabolic Medicine, Sapporo Medical University School of Medicine, Sapporo 060-8556, Japan; 3Department of Cellular Physiology and Signal Transduction, Sapporo Medical University School of Medicine, Sapporo 060-8556, Japan

**Keywords:** 3D spheroid culture, ARPE19, tamsulosin, α1 antagonist

## Abstract

The purpose of the present study was to examine the effect of the selective α1 antagonist tamsulosin (TAM) on human retinal pigment epithelium cells, ARPE 19. Two-dimension (2D) and three-dimension (3D) cultured ARPE 19 cells were used in the following characterizations: (1) ultrastructure by scanning electron microscopy (SEM) (2D); (2) barrier functions by transepithelial electrical resistance (TEER) measurements, and FITC-dextran permeability (2D); (3) real time cellular metabolisms by Seahorse Bioanalyzer (2D); (4) physical properties, size and stiffness measurements (3D); and (5) expression of extracellular matrix (ECM) proteins, including collagen1 (COL1), COL4, COL6 and fibronectin (FN) by qPCR and immunohistochemistry (2D and 3D). TAM induced significant effects including: (1) alteration of the localization of the ECM deposits; (2) increase and decrease of the TEER values and FITC-dextran permeability, respectively; (3) energy shift from glycolysis into mitochondrial oxidative phosphorylation (OXPHOS); (4) large and stiffened 3D spheroids; and (5) down-regulations of the mRNA expressions and immune labeling of most ECM proteins in a concentration-dependent manner. However, in some ECM proteins, COL1 and COL6, their immunolabeling intensities were increased at the lowest concentration (1 μM) of TAM. Such a discrepancy between the gene expressions and immunolabeling of ECM proteins may support alterations of ECM localizations as observed by SEM. The findings reported herein indicate that the selective α1 antagonist, TAM, significantly influenced ECM production and distribution as well as cellular metabolism levels in a concentration-dependent manner.

## 1. Introduction

α1 adrenergic receptors (ARs), a heterogeneous family of receptors, are well recognized as playing significant roles in the regulation of the sympathetic regulatory system. So far, three α1 AR subtypes, α1A, α1B and α1D AR, have been identified in many species [1,2,3,4,5,6]. In addition, splice variants of the α1A AR subtype were also found in humans (α1A HSA.1-, 2-, 3- and 4-AR) [7,8] and in rabbits (α1A OCU.1-, 2- and 3-AR) [9]. Within ocular tissues, it has been revealed that α1A AR, or α1B AR, are predominantly expressed within the iris, choroid and retina or the ciliary body in the rabbit eye [10], and in fact, α1A ARs act as pivotal regulatory mechanisms for iris dilator muscle contraction [11,12], intraocular pressure (IOP) homeostasis [13,14,15,16,17] and corneal endothelial cell functions [18]. However, as of this writing, the contributions of α1A ARs are not well understood in terms of ocular pathophysiology, especially in posterior segments such as the retina, retinal pigment epithelium (RPE) and choroid.

Tamsulosin (TAM), a selective antagonist of the α1 AR, commonly used to treat urinary tract stones and dysuria associated with benign prostatic hyperplasia [19], has been shown to induce unfavorable ocular side effects called intraoperative floppy iris syndrome (IFIS) during cataract surgery [20,21,22]. As the clinical manifestations of IFIS, progressive miosis, iris waving and iris prolapse are frequently observed [23,24]. Since α1A AR is predominantly expressed within RPE [25], which is categorized within uveal tissues as similar to the iris and choroid, in which α1A AR is also expressed [10], we rationally speculated that TAM may also influence some functions and morphology changes within RPE cells. If such unknown TAM-induced effects on RPE occur, it would be of great interest to examine the drug-induced effects of TAM on RPE cells under physiological as well as pathological conditions.

Therefore, in the current study, as an initial step to elucidating possible unidentified TAM-induced effects on RPE under physiological conditions, two-dimension (2D) and three dimension (3D) cell cultures of the human retinal pigment epithelium cell line, ARPE 19 cells, were used and those were subjected to the following analyses: ultrastructure by scanning electron microscope (2D); barrier functions by transepithelial electron resistance (TEER) and FITC-dextran permeability (2D); real time cellular function by Seahorse Bioanalyzer (2D); physical properties (3D); and ECM protein expressions by qPCR and immunocytochemistry (2D and 3D).

## 2. Materials and Methods

### 2.1. 2D Culture of ARPE 19 Cells

All experiments using human derived cells were conducted in compliance with the tenets of the Declaration of Helsinki after approval by the internal review board of Sapporo Medical University. A commercially available human retinal pigment epithelium cell line, ARPE 19, was purchased from the American Type Culture Collection (ATCC, #CRL-2302™, certification from company is attached in the Appendix A) and cultured in 150 mm 2D culture dishes until they reached 90% confluence at 37 °C in 2D growing medium composed of HG-DMEM containing 10% FBS, 1% L-glutamine, 1% antibiotic–antimycotic, and the cultures were maintained by changing the medium every other day. For study of the drug-induced effects by tamsulosin (TAM) (Tokyo Chemical Industry, Tokyo, Japan), the 2D cultures of ARPE 19 cells were processed during Day 1 through 5 in the absence or presence of 1, 10 or 100 μM TAM, of which the concentration levels were as described in a previous study in which neural cells were used [26].

### 2.2. Scanning Electron Microscopy (SEM) Analysis, Transepithelial Electron Resistance (TEER) and FITC-Dextran Permeability Measurements of 2D Cultured ARPE 19 Cell Monolayer

ARPE 19 cell monolayers were cultured in a TEER plate (0.4 μm pore size and 12 mm diameter; Corning Transwell, Sigma-Aldrich, St. Louis, MA, USA) and analyzed by (1) scanning electron microscopy using HITACHI S-4300 microscope operated at 5 keV (the detector features 1280 × 960 pixel), (2) the TEER values (Ωcm^2^) using an electrical resistance system (KANTO CHEMICAL CO. INC., Tokyo, Japan) and (3) FITC-dextran permeability measurements by measuring the fluorescence intensity of the amount of FITC that permeated through the membrane from the basal compartment to the apical compartment during a period of 60 min as described in a previous study [27,28,29]. 

### 2.3. Measurement of Real-Time Cellular Metabolic Functions of 2D ARPE 19 Cells

The rates of oxygen consumption (OCR) and extracellular acidification (ECAR) of 2D cultured HRPE cells in the absence and presence of 1, 10 or 100 μM TAM were measured using a Seahorse XFe96 Bioanalyzer (Agilent Technologies, Santa Clara, CA, USA) according to the manufacturer’s instructions. Briefly, approximately 20 × 10^3^ of 2D cultured cells were each placed in a well of a XFe96 Cell Culture Microplate (Agilent Technologies, CA, USA, #103794-100). Following centrifugation of the plate at 1600× *g* for 10 min, the culture medium was replaced with 180 μL of assay buffer (Seahorse XF DMEM assay medium (pH 7.4, Agilent Technologies, #103575-100), supplemented with 5.5 mM glucose, 2.0 mM glutamine and 1.0 mM sodium pyruvate). The assay plates were incubated in a CO_2_-free incubator at 37 °C for 1 hour prior to the measurements. OCR and ECAR were simultaneously measured using the Seahorse XFe96 Bioanalyzer under 3 min mix and 3 min measure protocols at baseline and following the injection of oligomycin (final concentration: 2.0 μM), carbonyl cyanide p-trifluoromethoxyphenylhydrazone (FCCP, final concentration: 5.0 μM), a mixture of rotenone/antimycin A (final concentration: 1.0 μM) and 2-deoxyglucose (2-DG, final concentration: 10 mM). Spare Respiratory Reserve was determined by the difference between the baseline OCR and those supplemented with FCCP. Glycolytic Reserve was determined by the difference in ECAR after the addition of oligomycin.

### 2.4. Preparation of 3D ARPE 19 Spheroids

The 2D cultured ARPE 19 cells prepared as above were further processed for 3D spheroid preparation by a method described recently [30,31]. Briefly, 2D cultured ARPE-19 were suspended in spheroid medium composed of 2D growth medium supplemented with 0.25% methylcellulose (Methocel A4M) to facilitate stable 3D spheroid morphology. Approximately 20,000 ARPE 19 cells/28 μL of spheroid medium were placed into each well of the hanging drop culture plate (# HDP1385, Sigma-Aldrich) at Day 0, and thereafter half of the medium was replaced on each following day until Day 5. As shown in Appendix A, the 3D ARPE 19 spheroid became a down-sized and matured form during the five days culture. For studying drug-induced effects of TAM at different concentrations (0, 1, 10 or 100 μM), TAM was added to the spheroid medium to maintain these concentrations during Days 1 through 5, as above. 

To evaluate physical properties of the 3D ARPE 19 spheroids, the mean size and the physical stiffness were determined as described recently [30]. Briefly, the mean size was determined by measuring the largest cross-sectional area (CSA) of the phase contrast images (PC, Nikon ECLIPSE TS2; Tokyo, Japan) using the Image-J software version 1.51n (National Institutes of Health, Bethesda, MD, USA). For the physical stiffness, the force required (μN) to compress a single living 3D spheroid to its semidiameter (μm) during 20 sec was measured using a micro-squeezer (MicroSquisher, CellScale, Waterloo, ON, Canada), and force/displacement (μN/μm) was calculated [30]. 

### 2.5. Immunocytochemistry of 2D ARPE 19 Cells and 3D ARPE 19 Cells Spheroids

Immunocytochemistry of the 2D and 3D cultured ARPE 19 cells was processed as described previously, with minor modifications [31,32]. All procedures were performed at room temperature unless otherwise stated. Briefly, 2D and 3D cultured ARPE 19 cells were fixed in 4% paraformaldehyde in PBS overnight, blocked in 3% BSA in PBS for 3 h, washed twice with PBS for 30 min, and thereafter they were sequentially treated with (1) an anti-human COL1, COL4, COL6 or FN rabbit antibody (1:200 dilutions) at 4 °C overnight; (2) washing three times with PBS for 1 h each; (3) 1:1000 dilutions of a goat anti-rabbit IgG (488 nm), phalloidin (594 nm) and DAPI for 3 h; and (4) mounting ProLong Gold Antifade Mountant with a cover glass. Immunofluorescent serials-axis images with a 2.2 μm interval at 35 μm height from their surface were obtained with a Nikon A1 confocal microscope using a ×20 air objective with a resolution of 1024 × 1024 pixels. 

### 2.6. Other Analytical Methods

Quantitative PCR analysis using specific primers and Taqman probes (Appendix A) and all statistical analyses using Graph Pad Prism 9 (GraphPad Software, San Diego, CA, USA) were performed as described previously [30].

## 3. Results

To study the drug-induced effects of the selective α1 antagonist tamsulosin (TAM) on RPE cells under physiological conditions, 2D cultured ARPE 19 cells were subjected to the following analyses: (1) ultra-structure determination by scanning electronic microscopy (SEM); (2) barrier functions of their 2D monolayers by transepithelial electron resistance (TEER) measurements and FITC-dextran permeability; (3) the expression of ECM proteins, including COL1, COL4, COL6 and FN; and (4) real time cellular metabolism analyses using a Seahorse XFe96 Bioanalyzer. As shown in Figure 1A, SEM revealed that dense ECM protein deposits spread all over the surface of the ARPE monolayers in the absence of TAM. However, in contrast, TAM significantly altered the localizations of the ECM proteins in a concentration-dependent manner; that is, there were many substantially enlarged ECM deposits located on the surface sparsely covered by ECM proteins. Consistent with these SEM observations, TAM induced a substantial increase of the barrier functions, that is, the increase of the TEER values and decrease of the FITC-dextran permeability of the 2D ARPE 19 monolayers (Figure 1B,C). In addition, those observations were rationally supported by a qPCR analysis (Figure 2) and immunocytochemistry findings (Appendix A) concerning major ECM proteins; namely, the expression of COL1, COL4, and FN or COL6 were significantly or relatively down-regulated by TAM in a concentration-dependent manner. Furthermore, a Seahorse real time cellular metabolic analysis of our prepared 2D cultured ARPE 19 cells showed acceptable responses, suggesting that the biological states of these cells were quite healthy, and that TAM induced an energy shift from glycolysis to mitochondrial oxidative phosphorylation (OXPHOS) (Figure 3), as evidenced by a concentration-dependent increase in the Spare Respiratory Reserve of the OCR and a decrease in the Glycolytic Reserve of the ECAR in the presence of TAM. Therefore, these results indicated that TAM caused significant effects on the structure and functions of 2D cultured ARPE 19 cells in a concentration-dependent manner.

Since it is well known that the uveal structures, including the iris, ciliary body and choroid, are not simple cell monolayer structures of RPE, this suggests that 3D culture models will be required to develop these related research fields [33] as well as the pathologic conditions of RPE such as proliferative vitreoretinopathy (PVR) [34]. Therefore, in preparation for future research studies, the drug-induced effects of TAM on the physical properties, size and stiffness, and the expression of ECM proteins of the 3D ARPE 19 spheroids, which are generally considered to be a more representative model for replicating organs [35], were investigated. As shown in Figure 4, Appendix A, TAM induced significant enlargement and hardening of the 3D ARPE 19 spheroids in a concentration-dependent manner. In addition, the mRNA expression of all four ECM proteins was significantly decreased upon the administration of TAM in a concentration-dependent manner as observed in the 2D ARPE 19 cells (Figure 5). However, in contrast, immunocytochemistry analysis indicated that expressions of COL1 and COL6 were increased in the presence of 1 μM TAM as compared with the non-treated control, although those of all ECM proteins were decreased with increasing TAM concentrations (Figure 6). Taken together, these results suggested that the discrepancy between qPCR and immunocytochemistry may be caused by TAM-induced alteration of the distributions of the ECM proteins as observed in the 2D ARPE 19 cells, resulting in the characteristic changes of the physical properties of the 3D ARPE 19 spheroids, as above.

## 4. Discussion

The lower urinary tract symptoms caused by benign prostatic hypertrophy (BPH) are well known and recognized as the most frequent urologic conditions in older men, and they are usually treated by α1 AR antagonists (α-1ARAs) [36,37]. Among the several α1ARAs, TAM is most commonly used because of the fewer adverse effects as compared with other α1ARAs, such as terazosin and doxazosin [38]. However, to the contrary, previous studies have demonstrated that the risk for IFIS is particularly higher in the TAM user as compared with users of other α1ARAs during cataract surgery [23,39,40,41]. IFIS increases the risk of serious complications during cataract surgery, particularly if surgeons are unaware of the use of α1ARAs [42]. In addition, it is known that α1AR is also expressed within RPE [10], so we reasonably speculated that the presence of the unidentified TAM induced some effects on RPE. In fact (and quite interestingly), it has been suggested that TAM may induce some favorable effects in RPE cells in diabetic retinopathy (DR) based on a recent study reporting that TAM may induce beneficial effects in diabetic nephropathy (DN) [43]. These conclusions were supported by the following results: (1) TAM reduced a high glucose-induced expression of TNF-α, IL-6, IL-8, MMP-2 and MMP-9; (2) TAM inhibited the expression of VCAM-1 and ICAM-1, and a high glucose-induced expression of fibrosis factors such as COL-1 and TGF-β1; and (3) TAM reduced oxidative stress by inhibiting the generation of ROS, thus preventing the activation of p38. In addition, therapy involving the use of a combination of drugs that regulate G protein couple receptor (GPCR) signaling pathways including TAM was found to beneficially inhibit the development of early diabetic retinopathy [44] as well as retinal degeneration [45,46], as compared with the use of each drug individually. In the current study, using 2D and 3D cultures of ARPE 19 cells, we were able to successfully evaluate the drug-induced effects of TAM and obtained the following results: TAM significantly altered the distribution of ECM deposits and the energy balance between glycolysis and OXPHOS, and the increased barrier functions in the 2D ARPE 19 monolayers, and large and stiffer 3D ARPE 19 spheroids, and these effects were concentration dependent. Therefore, considering the collective findings reported herein, it appears that TAM not only modulates the biological activities of RPE cells, but also may potentially become a therapeutic target for the treatment of RPE-affected disorders, although some TAM-induced ocular adverse risks need to be taken into consideration, in addition to IFIS. In fact, it was reported that TAM induced an increase in choroidal thickness [47] and choroidal detachment [48]. 

So far, it is postulated that α1ARAs affect the iris dilator muscle through α1AR causing irreversible atrophy of the iris dilator muscle because pre-operative cessation of α1ARAs does not decrease the risk of IFIS [22,23,49,50]. In fact, several in vitro studies have shown that α1AR is expressed within the iris dilator muscle in rats [51] and rabbits [52]. However, possible mechanisms of antagonism of α1AR by TAM within the iris dilator muscle have not been fully identified yet, although previous in vitro studies in rabbits indicated that TAM binds to iris melanin and, in turn, inhibits dilator muscles [53,54]. Since, for studying this issue further, some in vitro models replicating IFIS etiology will be required, our current study using 2D and 3D cultures of ARPE 19 cells may also be applicable for this study purpose. In contrast to RPE, the structure of the iris is very complicated, that is, the iris is composed of several types of cells including IPE, dilator and sphincter muscle, melanocytes and others [55], although both RPE and iris are categorized within uveal tissues epithelium, and α1AR is present in both tissues [25]. Therefore, to establish in vitro models replicating IFIS, 3D organoid culture will be required rather than 3D spheroid culture using iris derived cells.

As of this writing, despite our current insufficient understanding of the pathophysiological roles of α1AR and α1ARA within ocular tissues, our developed research strategy using newly developed 3D spheroid cultures in addition to the conventional 2D cell cultures represents a promising approach in this research field. However, as study limitations in the current investigation, no in vitro studies using RPE cells have appeared as far we know; therefore, we used 1–100 μM concentrations of TAM as was reported in a previous study of the TAM-induced effects on neuronal cells [26], which are anatomically similar to retinal cells. Nevertheless, in terms of the intraocular levels of TAM, previous studies by liquid chromatography-electrospray ionization tandem mass spectrometry indicate that the concentrations of TAM in aqueous humor and serum specimens were much lower, 0.1–4.7 ng/mL (2.4–11.5 nM) and 0.1–19.3 ng/mL (2.4–47.3 nM), respectively [22,56], as compared to the concentrations used in the current study (1–100 μM). It has also been shown that the AH and vitreous levels of drug concentrations were significantly different in topical versus systemic administration. That is, those levels were higher in AH than vitreous in the case of topically administered drugs [57], but, in contrast, vitreous levels were comparable or even higher than AH in the case of systemically administered drugs [58,59] or serum derived factors [60]. In addition, TAM-induced effects on RPE cells would also be expected to be different between short-term and long-term exposure, even when the same concentrations were used. Therefore, to develop a better understanding of the TAM-induced effects on RPE cells, we plan to perform additional experiments including measurements of cell growth, migration ability and related issues under a wider range of concentrations and different exposure periods, as well as using pathological states of RPE, as our next project.

## 5. Conclusions

In conclusion, as an initial step to investigate TAM-induced effects on RPE cells, and using 3D ARPE 19 spheroid cultures in addition to the conventionally used 2D cell cultures under physiological conditions, we found that TAM significantly modulated ECM expression and distribution as well as cellular metabolism states and that this modulation was concentration dependent.

## Figures and Tables

**Figure 1 bioengineering-09-00556-f001:**
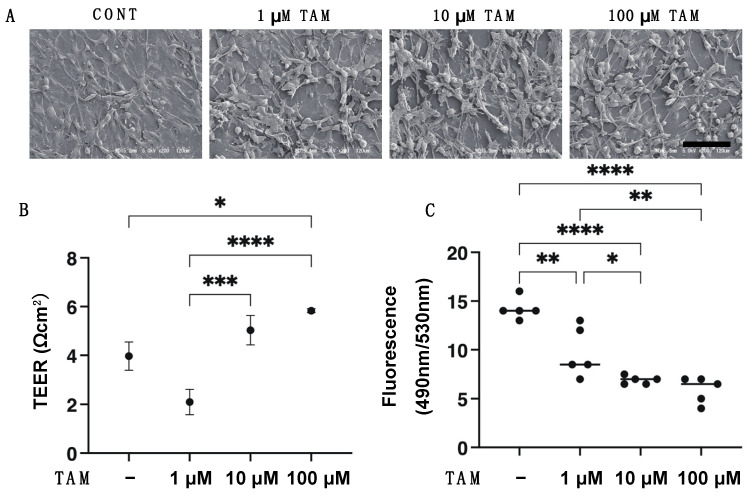
Effects of different concentrations of tamsulosin (TAM) on (**A**) ultrastructure by scanning electron microscopy (SEM), (**B**) transepithelial electrical resistance (TEER) and (**C**) FITC-dextran permeability of an ARPE 19 monolayer. Ultra-structure and barrier functions of an ARPE 19 cells monolayer obtained by their 2D culture at Day 5 in the absence or presence of 1 μM, 10 μM or 100 μM tamsulosin (TAM) were analyzed. Representative images by scanning electron microscopy (SEM, scale bar; 100 μm) are shown in panel A, and transepithelial electrical resistance (TEER) values and FITC-dextran permeability were plotted at panels (**B**,**C**), respectively. All experiments were performed in duplicate using fresh preparations (*n* = 5). Data are presented as arithmetic means ± standard error of the mean (SEM). * *p* < 0.05, ** *p* < 0.01, *** *p* < 0.005, **** *p* < 0.001 (ANOVA followed by a Tukey’s multiple comparison test).

**Figure 2 bioengineering-09-00556-f002:**
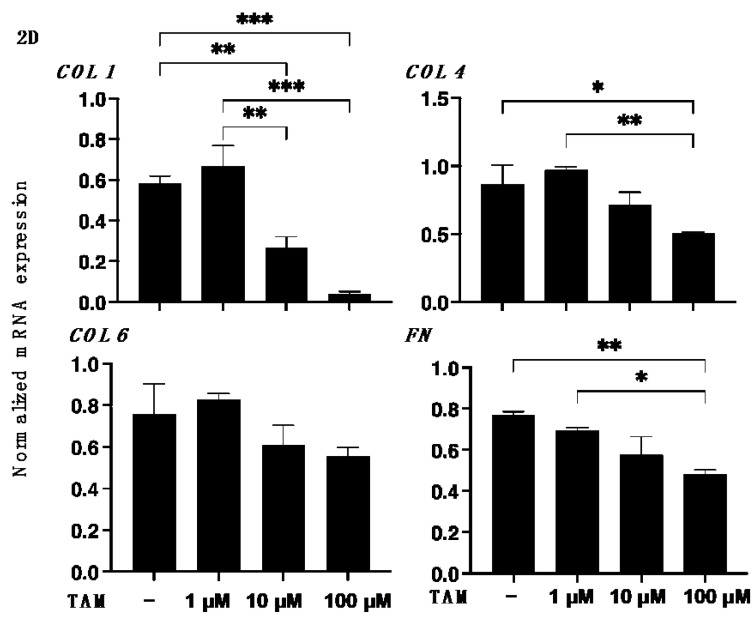
Effects of different concentrations of tamsulosin (TAM) on the mRNA expression of ECM proteins of the 2D cultured ARPE 19 cells. 2D cultured ARPE 19 cells at Day 5 in the absence or presence of 1 μM, 10 μM or 100 μM tamsulosin (TAM) were subjected to qPCR analyses to estimate the mRNA expression of ECM proteins including COL1, COL4, COL6 and FN. All experiments were performed in duplicate using fresh preparations (*n* = 5). Data are presented as the arithmetic mean ± standard error of the mean (SEM). * *p* < 0.05, ** *p* < 0.01, *** *p* < 0.005 (ANOVA followed by a Tukey’s multiple comparison test).

**Figure 3 bioengineering-09-00556-f003:**
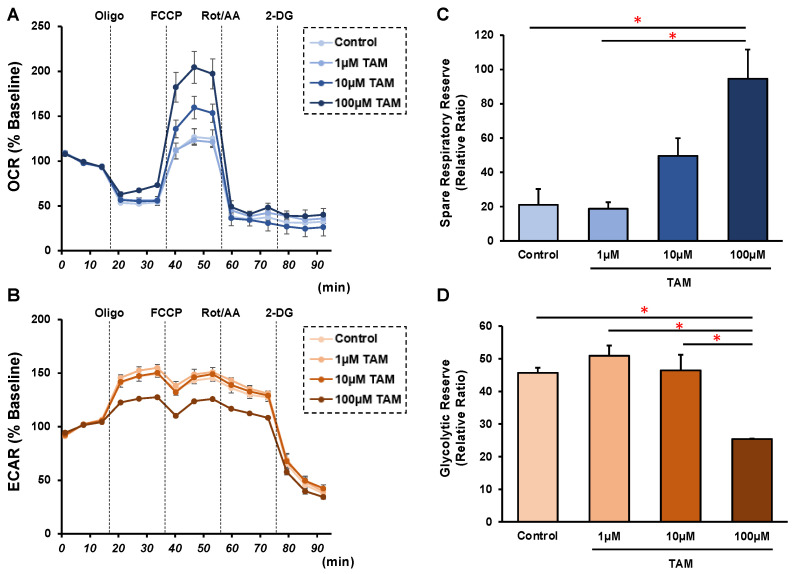
Effects of different concentrations of tamsulosin (TAM) on the cellular metabolic phenotype of the 2D cultured ARPE 19 cells. 2D cultured ARPE 19 cells at Day 5 were prepared in the absence or presence of 1 μM, 10 μM or 100 μM tamsulosin (TAM), and each sample was subjected to a real-time metabolic function analysis using a Seahorse XFe96 Bioanalyzer. Measurements of oxygen consumption rate (OCR, panel (**A**)) and extracellular acidification rate (ECAR, panel (**B**)) before drug injections (at baseline) were represented as 100% and their changes were determined by the following injections: oligomycin (a complex V inhibitor), FCCP (a protonphore), rotenone/antimycin (complex I/III inhibitors), and 2-DG (a hexokinase inhibitor). Relative ratios of the Spare respiratory Reserve and Glycolytic Reserve are plotted in panel (**C**,**D**), respectively. Oligo = oligomycin, Rot/AA = rotenone/antimycin A, 2-DG = 2-deoxyglucose. Fresh preparations were used in all experiments (*n* = 3). Data are presented as the mean ± the standard error of the mean (SEM). * *p* < 0.05 (ANOVA followed by a Tukey’s multiple comparison test).

**Figure 4 bioengineering-09-00556-f004:**
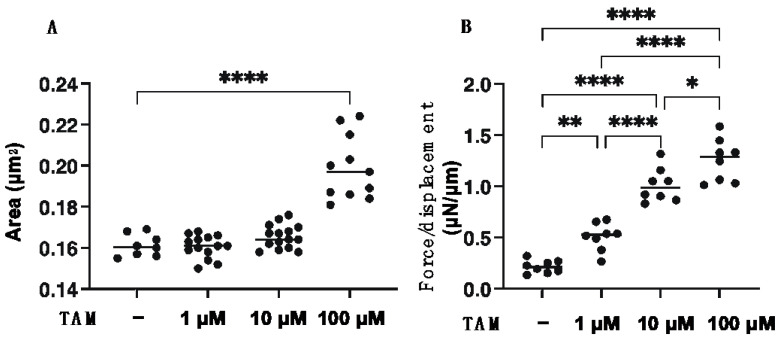
Effects of different concentrations of tamsulosin (TAM) on the physical properties, mean sizes and physical stiffness of the 3D ARPE 19 spheroids. The 3D ARPE 19 spheroids at Day 5 were prepared in the absence or presence of 1 μM, 10 μM or 100 μM tamsulosin (TAM). In the panel (**A**), their mean sizes and physical stiffness obtained by measuring the phase contrast images were plotted. In the panel (**B**), their physical stiffness was evaluated by the compressing them into their semidiameter (μm) during 20 sec using a micro-squeezer, and the requiring force/displacement (μN/μm) values were plotted. All experiments were performed in duplicate using fresh preparations (*n* = 12 and 15–20 for size and stiffness measurement, respectively). Data are presented as the arithmetic mean ± standard error of the mean (SEM). * *p* < 0.05, ** *p* < 0.01, **** *p* < 0.001 (ANOVA followed by a Tukey’s multiple comparison test).

**Figure 5 bioengineering-09-00556-f005:**
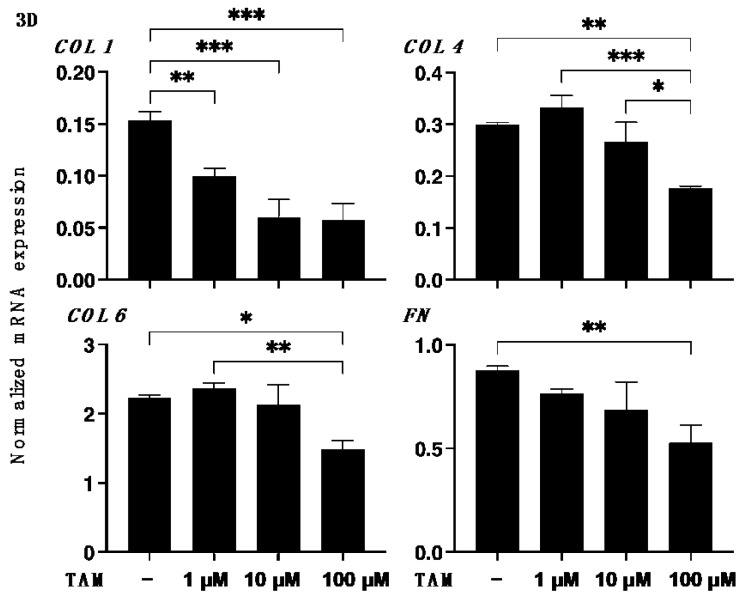
Effects of different concentrations of tamsulosin (TAM) on mRNA expression of ECMs in 3D spheroids of ARPE 19 cells. 3D ARPE 19 spheroids at Day 5 in the absence or presence of 1 μM, 10 μM or 100 μM tamsulosin (TAM) were subjected to qPCR analysis to estimate their mRNA expression of ECM proteins including COL1, COL4, COL6 and FN were performed. All experiments were performed in duplicate using fresh preparations (*n* = 10–15, each). Data are presented as arithmetic means ± standard error of the mean (SEM). * *p* < 0.05, ** *p* < 0.01, *** *p* < 0.005 (ANOVA followed by a Tukey’s multiple comparison test).

**Figure 6 bioengineering-09-00556-f006:**
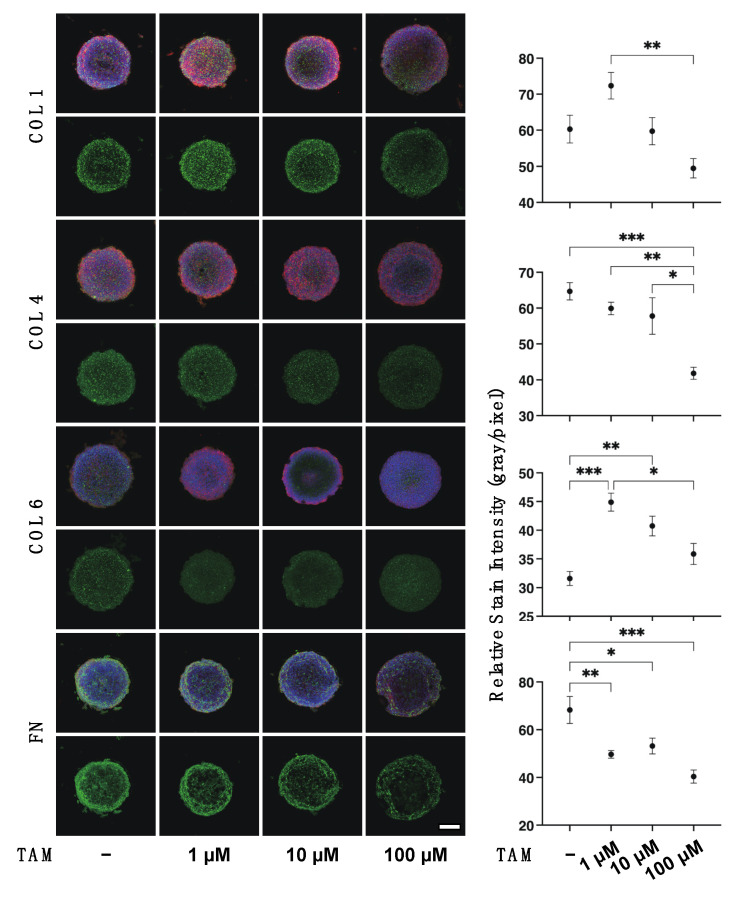
Representative confocal images showing the expression of ECMs in 3D ARPE 19 spheroids under several conditions. 3D ARPE 19 spheroids at Day 5 in the absence or presence of 1 μM, 10 μM or 100 μM tamsulosin (TAM) were subjected to immunohistochemistry analysis. Representative immunolabeling images by specific antibodies against collagen 1 (COL 1), collagen 4 (COL 4), collagen 6 (COL6), or fibronectin (FN) (green), DAPI (blue) and Phalloidin (red) are shown in panel A (scale bar: 100 μm). The staining intensities of the labeling of each ECM proteins were plotted in panel B. All experiments were performed in duplicate using fresh preparations consisting of 5 spheroids each. Data are presented as the arithmetic mean ± standard error of the mean (SEM). * *p* < 0.05, ** *p* < 0.01, *** *p* < 0.005 (ANOVA followed by a Tukey’s multiple comparison test).

## Data Availability

Not applicable.

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
