# Peer review of "The Selective α1 Antagonist Tamsulosin Alters ECM Distributions and Cellular Metabolic Functions of ARPE 19 Cells in a Concentration-Dependent Manner"

_bioengineering, 2022, doi:10.3390/bioengineering9100556_

Round 1

Reviewer 1 Report

The authors examine the effect of the selective a1 antagonist, tamsulosin (TAM) on the human retinal pigment epithelium cells using Two-dimension (2D) three-dimension (3D) cultured ARPE 19 cells. They established a new three-dimensional culture method and also observed the effect of TAM on RPE cells, including cell morphology, cell ECM location, and some expression changes in collagen molecules.

Major concerns

1. The apparent problem is that the authors have not elucidated why they studied the drug’s effect on RPE cells. Because the drug introduced by the author in the introduction can cause iris-related diseases in patients with benign prostatic hyperplasia, although it is expressed in RPE, does it cause RPE-related ophthalmic diseases?

2. Why didn’t the authors study iris cells? Has anyone checked it already? It is not clearly explained in the introduction.

3. The authors only examined the ECM distribution of RPE cells. If we want to see the effect of this drug on RPE cells, we should conduct a comprehensive test. For example, did the author test the impact of this drug on cell growth and migration ability?

4. Another question is, what is the basis for the author to choose the concentration of this drug? Because 1 mM, 10 mM, or 100 mM can be seen in the experimental results, it does not show a complete dose-dependent manner in all the tests, and the author did not explain it in the discussion

Author Response

The authors examine the effect of the selective a1 antagonist, tamsulosin (TAM) on the human retinal pigment epithelium cells using Two-dimension (2D) three-dimension (3D) cultured ARPE 19 cells. They established a new three-dimensional culture method and also observed the effect of TAM on RPE cells, including cell morphology, cell ECM location, and some expression changes in collagen molecules.

Major concerns

  1. The apparent problem is that the authors have not elucidated why they studied the drug’s effect on RPE cells. Because the drug introduced by the author in the introduction can cause iris-related diseases in patients with benign prostatic hyperplasia, although it is expressed in RPE, does it cause RPE-related ophthalmic diseases?
  2. Why didn’t the authors study iris cells? Has anyone checked it already? It is not clearly explained in the introduction.

Answer for #1 and #2; Thank you very much for this comment. We agree with the comment that the rationale for using TAM toward RPE rather than iris is a concern. To present this case in a more rational manner, additional information related to TAM and diabetic and retinal degeneration models were added in the Introduction section, as follows; “a1 adrenergic receptors (Ars), a heterogeneous family of receptors, are well recognized to play significant roles in the regulation of the sympathetic regulatory system. So far, three a1 AR subtypes, a1A, a1B and a1D AR, have been identified in many species [1-6]. In addition, splice variants of the a1A AR subtype were also found in humans; a1A has.1-, 2-, 3- and 4-AR [7, 8] and in rabbits; a1A OCU.1-, 2- and 3-AR [9]. Within ocular tissues, it has been revealed that a1A AR, or a1B AR were predominantly expressed within the iris, choroid and retina or the ciliary body in the rabbit eye [10], and in fact, a1a ARs play pivotal regulatory mechanisms of the iris dilator muscle contraction [11, 12], intraocular pressure (IOP) homeostasis [13-17], and corneal endothelial cells functions [18].However, as of this writing, the contribution of a1A ARs are not well understood in terms of ocular pathophysiology, especially in posterior segments, such as the retina, retinal pigment epithelium (RPE), and choroid.

Tamsulosin (TAM), a selective antagonist of the a1 AR, commonly used to treat urinary tract stones and dysuria associated with benign prostatic hyperplasia [19] has been shown to induced unfavorable ocular side effects called intraoperative floppy iris syndrome (IFIS) during cataract surgery [20-22]. As the clinical manifestations of the IFIS, progressive miosis, iris waving and iris prolapse are frequently observed [23, 24]. Since a1A AR is predominantly expressed within RPE [25], categorized within uveal tissues as similar to the iris and choroid, in which a1A AR is also expressed [10], we rationally speculated that TAM may also influence some functions and morphology changes within RPE cells. If such unknown TAM-induced effects on RPE occur, it would be of great interest to examine the drug induced effects of TAM toward RPE cells under physiological as well as pathological conditions.

Therefore, in the current study, as an initial step to elucidating possible unidentified TAM induced effects toward RPE under physiological conditions, two-dimension (2D) and three dimension (3D) cell culture of the human retinal pigment epithelium cell line, ARPE19 cells were used and those were subjected to following analyses; ultrastructure by scanning electron microscope (2D), barrier functions by transepithelial electron resistance (TEER) and FITC-dextran permeability (2D), real time cellular function by Seahorse Bioanalyzer (2D), physical properties (3D) and ECM protein expressions by qPCR and immunocytochemistry (2D and 3D).”.

  1. The authors only examined the ECM distribution of RPE cells. If we want to see the effect of this drug on RPE cells, we should conduct a comprehensive test. For example, did the author test the impact of this drug on cell growth and migration ability?

Answer; Thank you for this comment and the excellent advice that additional experiments such as cell growth and migration ability would be desirable. However, although we did not perform these experiments, we assumed that TAM may not significantly affect either cell growth or migration ability based upon the following reasons; 1) we used A RPE 19 cells in which cell growth was not stimulated and cell migration by TGF-b, LPS and others, 2) TAM does not affect cell growth and migration ability in malignant tumor cells (Am J Cancer Res. 2020; 10(12): 4386–4398.), 3) we used 2D RPE cells that were already confluent, 4) as shown in Fig. 2, the presence of TAM induced an energy shift between OCR and ECAR in a concentration dependent manner, suggesting that TAM-induced effects may not be sufficiently drastic to cause significant changes in cell growth as well as cell migration even in the current study, and 5) as shown in Supplemental Fig. 1 and similar to other source of cells, the 3D HRPE spheroid grew into matured forms during the 5-day culture period in the absence or presence of TAM, suggesting that TAM may also not induce drastic effects on 3D spheroid maturation, although some significant differences were observed in the physical properties, size and stiffness and the ECM expression of the 3D HRPE spheroids by TAM. Similar alterations in the physical properties and ECM expressions of 3D spheroids were also observed in our previous studies using other sources of cells including orbital fibroblasts, 3T3-L1 cells, human trabecular meshwork cells, human conjunctival fibroblasts, in addition to HRPE cells. However, the suggested measurements will be needed in the case that we will study TAM induced effects toward RPE cells under pathogenic conditions as the next step of our investigation. Therefore, this information is included in the 3rd paragraph of the Discussion; “As of this writing, despite our current insufficient understanding of the pathophysiological roles of a1AR and a1ARA within ocular tissues, our developed research strategy using newly developed 3D spheroid cultures in addition to the conventional 2D cell cultures represents a promising approach in this research field. However, as study limitations in the current investigation, no in vitro studies using RPE cells have appeared as far we survey, and therefore, we used 1-100 mM concentrations of TAM as was reported in a previous study of the TAM induced effects toward neuronal cells [26], which are anatomically similar to retinal cells. Nevertheless, in terms of the intraocular levels of TAM, previous studies by liquid chromatography-electrospray ionization tandem mass spectrometry indicate that the concentrations of TAM in aqueous humor and serum specimens was much lower 0.1-4.7 ng/ml (2.4-11.5 nM) and 0.1-19.3 ng/ml (2.4-47.3 nM), respectively [22, 56] as compared to the concentrations used in the current study (1-100 mM). It has also been shown that the AH and vitreous levels of drug concentrations were significantly different between topical versus systemic administration. That is, those levels were higher in AH than vitreous in the case of topically administered drugs [57], but, in contrast, vitreous levels were comparable or even higher than AH in the case of systemically administering drugs [58, 59] or serum derived factors [60]. In addition, TAM induced effects on RPE cells would also be expected to be different between short-term and long-term exposure, even when the same concentrations were used. Therefore, to develop a better understanding of the TAM induced effects on RPE cells, we plan to perform additional experiments including measurements of cell growth, migration ability and related issues under wider ranges of concentrations and different exposure periods as well as using pathological states of RPE as our next project.”.

  1. Another question is, what is the basis for the author to choose the concentration of this drug? Because 1 mM, 10 mM, or 100 mM can be seen in the experimental results, it does not show a complete dose-dependent manner in all the tests, and the author did not explain it in the discussion

Answer; Thank you for this comment. Concerning the TAM concentrations used in the current experiments, we agree that much higher concentrations were used as compared to in vivo measurements. However, in the case of the systemic administered drug, the concentrations would be different between ate aqueous humor and vitreous, and the concentrations may be higher in the vitreous than in the aqueous humor. Therefore, the issue of the difference in the vitreous and aqueous humor concentrations and the rationale for using these concentrations is that in a previous study, we used neural cells (ref 26) which are anatomically similar to retina cells because no previous study using TAM and RPE have been reported. These issues are included as study limitations within the 3rd paragraph of Discussion as follows; “As of this writing, despite our current insufficient understanding of the pathophysiological roles of a1AR and a1ARA within ocular tissues, our developed research strategy using newly developed 3D spheroid cultures in addition to the conventional 2D cell cultures represents a promising approach in this research field. However, as study limitations in the current investigation, no in vitro studies using RPE cells have appeared as far we survey, and therefore, we used 1-100 mM concentrations of TAM as was reported in a previous study of the TAM induced effects toward neuronal cells [26], which are anatomically similar to retinal cells. Nevertheless, in terms of the intraocular levels of TAM, previous studies by liquid chromatography-electrospray ionization tandem mass spectrometry indicate that the concentrations of TAM in aqueous humor and serum specimens was much lower 0.1-4.7 ng/ml (2.4-11.5 nM) and 0.1-19.3 ng/ml (2.4-47.3 nM), respectively [22, 56] as compared to the concentrations used in the current study (1-100 mM). It has also been shown that the AH and vitreous levels of drug concentrations were significantly different between topical versus systemic administration. That is, those levels were higher in AH than vitreous in the case of topically administered drugs [57], but, in contrast, vitreous levels were comparable or even higher than AH in the case of systemically administering drugs [58, 59] or serum derived factors [60]. In addition, TAM induced effects on RPE cells would also be expected to be different between short-term and long-term exposure, even when the same concentrations were used. Therefore, to develop a better understanding of the TAM induced effects on RPE cells, we plan to perform additional experiments including measurements of cell growth, migration ability and related issues under wider ranges of concentrations and different exposure periods as well as using pathological states of RPE as our next project.”.

Reviewer 2 Report

Authors should show more data on RPE quality and its validation. It's hard to trust the conclusion if one doesn't know if the quality of cells is good enough. 

TEER is Transepithelial electrical resistance, not endothelial. 

3D model of RPE lacks some details and what is that 3D model brings that 2D cant and in-vivo RPE works as a monolayer, so I don't understand the use  of 3D model

Author Response

  1. Authors should show more data on RPE quality and its validation. It's hard to trust the conclusion if one doesn't know if the quality of cells is good enough.

Answer; Thank you for this comment. Concerning the quality of RPE cells, a certificate from the American Type Culture Collection (ATCC, #CRL-2302™) is attached within the supplemental material. In addition, our current real time cellular metabolic analysis the 2D cultured RPE cells by a Seahorse bioanalyzer indicated that the cellular responses were acceptable. Since it is well known that this measurement is an extremely sensitive analysis that is indicative of the health of living cells. We therefore conclude that the RPE cells used in this study were of good quality. This information is included in the 1st paragraph of Result; “To study the drug induced effects of the selective a1 antagonist, tamsulosin (TAM) toward RPE cells under physiological conditions, 2D cultured ARPE19 cells were subjected to the following analyses; 1) ultra-structure determination by scanning electronic microscopy (SEM), 2) barrier functions of their 2D monolayers by transepithelial electron resistance (TEER) measurements and FITC-dextran permeability, 3) the expression of ECM proteins, including COL1, COL4, COL6 and FN, and 4) real time cellular metabolism analyses using a Seahorse XFe96 Bioanalyzer. As shown in Fig 1A, SEM revealed that the dense ECM proteins deposits spread all over the surface of the ARPE monolayers in the absence of TAM. However, in contrast, TAM significantly altered the localizations of the ECM proteins in a concentration dependent manner, that is, there were lots of substantially enlarged ECM deposits located on the surface sparsely covered by ECM proteins. Consistently with these SEM observations, TAM induced substantial increase of the barrier functions, that is, the increase of the TEER values and decrease of the FITC-dextran permeability of the 2D ARPE19 monolayers (Fig. 1B and C). In addition, those observations were rationally supported by a qPCR analysis (Fig. 2) and immunocytochemistry findings (supplemental Fig. 1) concerning major ECM proteins, namely, the expression of COL1, COL4, and FN or COL6 were significantly or relatively down-regulated by TAM in a concentration dependent manner. Furthermore, a Seahorse real time cellular metabolic analysis of our prepared 2D cultures ARPE 19 cells showed acceptable responses, suggesting that the biological states of these cells were quite healthy, and that TAM induced an energy shift from glycolysis to mitochondrial oxidative phosphorylation (OXPHOS) (Fig. 3), as evidenced by a concentration-dependent increase in the Spare Respiratory Reserve of the OCR and a decrease in Glycolytic Reserve of the ECAR in the presence of TAM. Therefore, these results indicated that TAM caused significant effects toward structure and functions of 2D cultured ARPE 19 cells in a concentration dependent manner.”. 

  1. TEER is Transepithelial electrical resistance, not endothelial.

Answer; Thank you for this comment. As pointed out, TEER was corrected to” Transepithelial electrical resistance”, but not endothelial.

Reviewer 3 Report

- the authors analyzed mRNA expression of ECMs; however, the protein expression levels need to be quantified

- the discussion is too poor. the authors need to deep the biological significance of their data and compare them with molecular mechanisms already found in different cell models.

- how did the authors choose the concentration of TAM? please clarify

- the authors found that the minimum concentration of TAM have an opposite effect compared to increased doses; do the authors have an explanation for that?

Author Response

  1. the authors analyzed mRNA expression of ECMs; however, the protein expression levels need to be quantified.

Answer; Thank you for this comment. As suggested, in addition to the mRNA expression of ECMs, the immunocytochemistry of 2D cultured cells similar to 3D spheroids were included in the supplemental Fig. 1 to show their protein expression levels.

  1. the discussion is too poor. the authors need to deep the biological significance of their data and compare them with molecular mechanisms already found in different cell models.

Answer; Thank you for this comment. As suggested by several comments from 3 reviewers, we completely agree with that current study purpose and discussion were somewhat ambiguous. Therefore, the study purpose was revised based upon recent interesting studies suggesting that TAM induced effects toward RPE may be a therapeutic candidate for the treatment of retinal degeneration and diabetic retinopathy. According this, the entire manuscript story was revised and improved.

  1. how did the authors choose the concentration of TAM? please clarify.
  2. the authors found that the minimum concentration of TAM have an opposite effect compared to increased doses; do the authors have an explanation for that?

Answers for #3 and #4; Thank you for these comments. In terms of the TAM concentrations used in the current experiments, we agree with much higher concentrations were used as compared with in vivo measurements. However, in case of the systemic administered drug, the concentrations would clearly be different between aqueous humor and vitreous, and may be higher in the vitreous than in the aqueous humor. In addition, as suggested by other reviewers, the study rationale for using TAM toward RPE rather than the iris was ambiguous. Therefore, the issue of the difference of the vitreous and aqueous humor concentrations and rationale for using these concentrations as followed by previous study using neural cells which are anatomically similar to retina cells because no previous study using TAM and RPE were included within the the 3rd paragraph of the Discussion; “As of this writing, despite our current insufficient understanding of the pathophysiological roles of a1AR and a1ARA within ocular tissues, our developed research strategy using newly developed 3D spheroid cultures in addition to the conventional 2D cell cultures represents a promising approach in this research field. However, as study limitations in the current investigation, no in vitro studies using RPE cells have appeared as far we survey, and therefore, we used 1-100 mM concentrations of TAM as was reported in a previous study of the TAM induced effects toward neuronal cells [26], which are anatomically similar to retinal cells. Nevertheless, in terms of the intraocular levels of TAM, previous studies by liquid chromatography-electrospray ionization tandem mass spectrometry indicate that the concentrations of TAM in aqueous humor and serum specimens was much lower 0.1-4.7 ng/ml (2.4-11.5 nM) and 0.1-19.3 ng/ml (2.4-47.3 nM), respectively [22, 56] as compared to the concentrations used in the current study (1-100 mM). It has also been shown that the AH and vitreous levels of drug concentrations were significantly different between topical versus systemic administration. That is, those levels were higher in AH than vitreous in the case of topically administered drugs [57], but, in contrast, vitreous levels were comparable or even higher than AH in the case of systemically administering drugs [58, 59] or serum derived factors [60]. In addition, TAM induced effects on RPE cells would also be expected to be different between short-term and long-term exposure, even when the same concentrations were used. Therefore, to develop a better understanding of the TAM induced effects on RPE cells, we plan to perform additional experiments including measurements of cell growth, migration ability and related issues under wider ranges of concentrations and different exposure periods as well as using pathological states of RPE as our next project.”.

Round 2

Reviewer 1 Report

The author has answered the questions  and revised the article as appropriate

Reviewer 3 Report

the authors addressed all issues.